# Seeing the Finish Line? Retirement Perceptions and Wellbeing among Social Workers

**DOI:** 10.3390/ijerph17134722

**Published:** 2020-06-30

**Authors:** John Moriarty, Patricia Gillen, John Mallett, Jill Manthorpe, Heike Schröder, Paula McFadden

**Affiliations:** 1Centre for Evidence and Social Innovation, Queen’s University Belfast, Belfast BT96AX, UK; 2Southern Health and Social Care Trust, Rosedale, Moyallen Road, Gilford BT63 5JX, UK; patricia.gillen@southerntrust.hscni.net; 3Institute of Nursing and Health Research, Ulster University, Newtownabbey BT37 0QB, UK; 4School of Psychology, Ulster University, Coleraine BT52 1SA, UK; j.mallett@ulster.ac.uk; 5NIHR Policy Research Unit in Health and Social Care Workforce Research Unit, King’s College London, London WC2R 2LS, UK; jill.manthorpe@kcl.ac.uk; 6School of Management, Queen’s University Belfast, Belfast BT95EE, UK; h.schroder@qub.ac.uk; 7School of Applied Social and Policy Sciences, University of Ulster at Magee, Londonderry BT487JL, UK

**Keywords:** mental health, workplace, retirement, extended working life, organisational policy

## Abstract

Planning for future health and social services (HSS) workforces must be informed by an understanding of how workers view their work within the context of their life and the challenges they will face across the course of life. There is a range of policies and provisions that states and organisations can adopt to create sustainable careers, support wellbeing at work, and extend working lives where appropriate, but the potential impact of these policies on the make-up of the workforce remains under investigation. This paper makes the case that service planners need to appreciate complex interplay between wellbeing and career decisions when planning the future workforce. It makes use of a recent survey of United Kingdom (UK) social workers (*n* = 1434) to illustrate this interplay in two ways. First, we present the analysis of how social workers’ perception of retirement and extended working lives are associated with dimensions of Work-Related Quality of Life (WRQL). We find that social workers who agreed that a flexible working policy would encourage them to delay their retirement scored lower on the Home-Work Interface and Control at Work dimensions of WRQL, while social workers who indicated a perception that their employer would not wish them to work beyond a certain age had lower Job and Career Satisfaction scores. Second, we propose a new typology of retirement outlooks using latent class analysis of these attitudinal measures. An 8-class solution is proposed, and we demonstrate the predictive utility of this scheme. Results are discussed in terms of the challenges for ageing Western populations and the usefulness of analysis such as this in estimating the potential uptake and impact of age-friendly policies and provisions.

## 1. Introduction

Population ageing is changing how we think about the future provision of healthcare and allied social services. The United Nations now estimates that the world is on course to have more people over 65 than under 15 for the first time ever by 2074 [1]. In the United Kingdom, this crossover had already occurred in 2016, and by 2060, the population of over-65s and under-25 s are projected to be of approximately equal size.

These changing demographics will affect both the demand for health and social services (HSS) and the supply of workers to those services [2]. Incentivising older workers to defer retirement and continue working beyond the traditional period of pension entitlement is one policy approach being actively explored in many western countries, including the United Kingdom [3]. The purported benefit of extending working lives is that it maintains the size of the working-age population to the extent that the numbers retained compensate for declining fertility rates and fewer young people maturing into working age [3].

Furthermore, if policymakers assume that work supports better health, they may anticipate a double-dividend, whereby those who stay in work longer also remain healthier for longer and require less support from HSS over their later lives. However, this is to dramatically over-simplify both the relationship between work and wellbeing and the reasons for which people choose to retire [4]. Focusing on the concept of sustainable careers and responding to the call of De Vos et al. [5] to investigate careers in occupations that diverge from traditional white-collar professions, we investigate the dynamic interplay between individuals and their work context and how this relationship shapes the sustainability or unsustainability of careers in later life in the context of health and wellbeing among health and social services workers. Careers can be defined as “sequential employment-related experiences through time and across space” [6] (p. 67), and it is generally acknowledged that employment careers are intertwined with trajectories related to family life [7]. Early career models theorised the existence of predetermined age-graded life and career stages that, for example, foresaw a clear transition into retirement at a specific biological age, which was generally associated with and in response to physical and psychological decline (see, for example, Super [8]). More current conceptualisations of careers break with the assumption that biological age determines levels of ability, work engagement and productivity. This suggests that (late) career transitions and experiences are shaped by a multitude of factors that differ from individual to individual, depending on the context as well as over time [5,9,10].

We, therefore, assume that the experiences of HSS workers differ and that the differences in career trajectories and expectations, coupled with differences in employment and family context, shape their late employment careers and retirement transitions as well as assumptions about career sustainability. Many HSS workers already work beyond the point at which their work is health-enhancing. Jobs that are physically or emotionally intense can create a strain that can accumulate over time into injury, incapacitation, or burnout [11,12]. This reduces these individuals’ ability to maintain a sustainable career in later life. Conversely, there are individuals for whom mandatory, normative, or expected retirement precipitates a phase of reduced activity, loss of purpose, and engagement, the harm of which outweighs any reprieve from daily pressures [13].

Retirement is a career transition that occurs in a complex context [3]. Pull factors into retirement can include the desire to devote more time to family and home life, synchronisation with a partner’s career and retirement plans, financial rewards (e.g., statutory benefits or private pension entitlements), or pursuit of alternative occupations or opportunities with reduced financial risk. Push factors include organisational stigma and age norms against old age and working into later adulthood. These comprise assumptions about older individuals’ lower productivity, health levels, and lack of willingness to learn and to embrace change [14]. Such age norms, in fact, influence older individuals’ recruitment and retention opportunities [15,16] as well as their access to promotions and job-related training [17]. Evidence suggests that older individuals internalise and reproduce such age norms [18]. Furthermore, research shows that older workers’ subjective perception that their line managements hold stereotypes against them might lead to their work disengagement [19]. Such a stereotype threat, hence, causes reduced psychological wellbeing, performance decrements, and higher turnover and/or retirement intent [20]. Dominant discourses that emphasise the need for older workers to “step aside” and “give a fair chance” to younger professionals [21] are often based on organisational cost-saving strategies by eventually replacing older, more experienced staff who are more highly paid with younger and less experienced staff at lower salary levels [3].

Recent estimates suggest that there are over 106,000 social workers employed in the United Kingdom, with one in five of those employed in adult services aged 55 and over, 19% aged 55–64 and 2% aged over 65 [22]. HSS professions in the UK have historically had high vacancy rates and turnover [23], with the child protection sector particularly susceptible to workforce volatility [24,25]. Concerns about stress and burnout and “presenteeism” have been repeatedly documented [12,24]. Furthermore, it is estimated that 24 percent of female and 21 percent of male HSS workers retire early on health grounds [26]. How service planners respond to the drive towards extended working lives and which career models are adopted to help individuals to stay in work longer requires careful thought. The key question is, “How will different workers respond to and be affected by different policy approaches?” For example, if a move to extend working lives will incur a cost to wellbeing for some HSS workers, policymakers should be interested in how many workers will be affected and to what extent.

This paper uses survey data from UK social workers to inform this area of policy. We present exploratory analysis in response to the question, “Are attitudes towards retirement and extended working lives related to wellbeing and quality of life?” Building from this, we test the evidence for a typology of workers derived from the varied perceptions of working in later life.

## 2. Methods

The paper draws on UK social workers’ responses to an online survey. The title of the survey was “Ageing at Work” and it aimed to capture attitudes among social work professionals towards ageing, career planning, and health and wellbeing. We refer to it herein as the Social Work Ageing Survey (SWAS). The survey was hosted by Ulster University and promoted by Community Care (www.communitycare.co.uk), whose subscribers receive regular updates on news related to social work and social care, and by the Northern Ireland Social Care Council, which regulates social services provision regionally. A total of 1434 participants engaged in the survey. Although all social workers were invited to respond, the sample skewed towards older workers, likely due to the subject matter. Only a minority of respondents reported having retired already (*n* = 70), reflecting the likelihood that most former social workers would have unsubscribed from professional newsletters.

### 2.1. Ethics

The survey research received approval by Ulster University’s Research Ethics Committee (RG3/26-Jan-18). No identifying information was collected, and participants were assured that their responses would be given anonymously, that they could withdraw from the survey at any point and that outputs would be managed such that no small groups would be described in detail. Data were stored and analysed on the secure servers of the participating universities.

### 2.2. Measures: Attitudes to Retirement and Work-Related Quality of Life

Work-Related Quality of Life (WRQL) [27] is a composite measure consisting of six subscales: Control at Work (CAW), depicting how far respondents feel they are involved in decisions that affect them at work (e.g., “I feel able to voice opinions and influence changes in my area of work”); General Well Being (GWB), depicting contentment with life; Home-Work Interface (HWI), capturing how much respondents agree that the organisation understands and tries to help them with pressures outside of work (e.g., “My employer provides adequate facilities and flexibility for me to fit work in around my family life”); Job Career Satisfaction (JCS) are respondents generally happy with their ability to do their job (e.g., “I have a clear set of goals and aims to enable me to do my job”); Stress at Work (SAW) asks questions on how much respondents feel they are experiencing stress at work; Working Conditions (WCS) asks respondents about the extent they agree they are happy with the conditions in which they work. Single responses to 24 statements are required on a 5-point Likert scale ranging from Strongly Disagree to Strongly Agree. In all cases, higher scores represent a higher quality of life, with the exception of Stress at Work, though this is reverse-coded such that a higher score represents lower stress in order to aggregate the scales into an overall WRQL. This recoded version is used throughout this paper. The subscales have been shown to have high reliability, with Cronbach’s alpha scores in a range from 0.75–0.86 [27].

The Intention to Leave scale [28] requires Likert responses from 1–5, indicating “never to always”, “very satisfying to totally dissatisfying”, or “highly unlikely to highly likely”. The questions asked include the following: (1) How often have you considered leaving your job? (2) How satisfying is your job in fulfilling your personal needs? (3) How often are you frustrated when not given the opportunity at work to achieve your personal work-related goals? (4) How often do you dream about getting another job that will better suit your personal needs? (5) How likely are you to accept another job at the same compensation level should it be offered to you? (6) How often do you look forward to another day at work? ITL has been reported as having a Cronbach’s alpha of 0.91 [28].

Questions on attitudes and plans around retirement were mainly adapted from the Attitudes to Working in Later Life module of the 2015 British Social Attitudes Survey [29]. Questions were adapted for survey responses and had categorical response options. Reasons for expecting to retire were suggested, and respondents asked to select as many options that apply to them. These options include (1) because you can receive state benefits; (2) because you can receive occupational or personal pension; (3) to retire at the same time as a partner; (4) because you can afford to; (5) because you want to; (6) because of ill-health; (7) because you do not expect your employer will allow you to work beyond this age; (8) some other reason. Respondents were asked to consider things their employer could do to help them to work longer and were given categorical responses and could select as many as applied to their circumstances. Options include (1) being able to retrain for a new role; (2) being offered a course to update your skills; (3) being able to work flexible hours; (4) being able to work part-time; (5) taking on a less demanding role; (6) having longer holidays; (7) being able to take a long break of a month or more.

*When do you expect to retire from your main job?* Responses were categorised into eight options from (1) in my 40 s, incrementally up to (5) planning to retire in my eighties, and included (7) not planning to retire, (8) no main job, (9) do not know, and (10) would rather not answer. Respondents were also asked if they expected to retire before, after, or at their anticipated pension age.

Career change questions were created for the current survey to measure the likelihood of changing careers and reasons why this was being considered. Respondents were asked if they planned any mid–late career changes using a 5-scale Likert response from extremely unlikely to extremely likely. They were asked to indicate all reasons for this planned mid-career change with reference to the following options, ticking as many of these that applied to their circumstances: (1) Because I want to have a variety of practice experiences; (2) because I find my job very stressful; (3) because I found my job was impacting on my health and wellbeing; (4) none of the above, I just wanted a change; (5) other reasons (box provided for narrative response).

### 2.3. Data Analysis

Ordinary least squares regressions were specified with each of these attitudinal indicators as a predictor of each dimension of WRQL. In Appendix A, we present the unadjusted coefficient with the attitude as a solo predictor, along with the adjusted coefficients from multivariate models with all attitudes entered together.

Latent class analysis (LCA) is a highly influential means of identifying unobserved groupings through patterning in observed responses [30]. Below, we first outline our priors to illustrate the type of classes that the SWAS data could be used to detect and the potential value of identifying these. We then describe the model specifications and measures used to extract latent groupings of social workers distinguished by their disposition to retirement and its potential impact on their health and wellbeing.

### 2.4. Prior Reasoning

From the perspective of organisational and state policymakers, we can consider the potential impacts at the individual level of policies that either extend working life or support a transition into retirement. In the preceding section, we highlighted workers’ needs to plan and achieve personal sustainability and that only some workers can achieve this by extending their careers. The following typology positions workers at different points relative to that point of sustainability and acknowledges that a given career pathway could represent sustainability for some workers and an unsustainable situation for others. This can be imagined through the combination of four variables. The main two axes are healthy versus unhealthy and retired versus career continued. These four groups can then be further divided by a hypothetical counterfactual, whereby the health status would be reversed if the retirement status were also reversed, e.g., retired social workers in poor health, whose health would be improved by continuing work. This increases the number of putative groups to eight. Finally, those counterfactuals in which health can be further divided by distinguishing those who would continue to work despite having formally retired from those continuing their career and maintaining their employment status. Thus the current workforce breaks down into the following six categories:
Healthy, in Work—Would be Unhealthy if RetiredHealthy, in Work—Would be the Same if Retired CompletelyHealthy, in Work—Would be the Same if Career Concluded, but Work ContinuedUnhealthy, in Work—Would be Healthy if Retired CompletelyUnhealthy, in Work—Would be Healthy if Career Concluded, but Work ContinuedUnhealthy, in Work—Would be the Same/Unhealthy if Retired

In this framework, policymakers would wish to maximise health and avoid people making transitions that worsen health. Therefore, distinguishing between different types of retirement circumstance is only valuable where that circumstance maintains or brings about a healthy outcome; hence, the final distinction is only applied to those for whom health status is linked to their work status. If such groups could be identified, described, and their relative size deduced, this would enable policymakers to conceptualise the types of transitions they wish to promote and avoid, the number of people at risk of harmful transitions and the magnitude of the potential adjustment of increased support for those nearing the normative retirement age. However, because these groups are constituted in part by a counterfactual which can never be directly observed, they would be difficult to enumerate and compare through combining survey items in a conventional contingency table. Nevertheless, the available SWAS survey data includes responses to valuable questions about hypothetical scenarios of retirement and of altered work provisions and may, therefore, be amenable to capturing hidden groups through latent class analysis (LCA).

With survey data from a majority working population, we accept that we are unlikely to be able to detect segmentation with only 70 retired respondents. Therefore, the six classes above plus a seventh heterogeneous class, Retired, were the groupings we attempted to detect through LCA.

### 2.5. Latent Class Model

We used the latent class analysis *lclass* option within Stata’s general structural equation modelling *gsem* framework [31,32]. This routine facilitates the inclusion of binary (logit), Likert (ologit) and continuous or scale (Gaussian) variables. We included the following measures in deriving classes: attitudes to retirement and mid-career change; age and stage at which individuals intend to retire; reasons for considering retirement; support for provisions to facilitate extended working life. The distribution of these variables can be found in Table 1, along with mean averages for Work-Related Quality of Life subscales.

The optimal fitting model was determined as that with the highest Aikike Information Criterion (AIC) score [30]. A plot of AIC scores for 4-class through to 14-class versions can be seen in Appendix A.

## 3. Results

### 3.1. Retirement Attitudes and Work-Related Quality of Life (WRQL)

Overall, regression output summarised in Appendix A shows there are several self-reported perceptions of retirement that correlate strongly with the dimensions of the Work-Related Quality of Life composite measure. Different indicators influence different aspects of quality of life, although there are no instances where scores on one dimension of wellbeing are significantly higher for a group of respondents, while another wellbeing dimension is significantly lower for the same group.

The largest association was seen for people who reported that they would consider retiring due to ill-health, who scored over 0.5 standard deviations lower on the composite WRQL scale. These individuals scored lower on all subscales, although the smallest association was for stress at work. Despite that stress scores were higher for the ill-health retirement group, this was not independently significant when adjusted for other retirement perceptions.

In the reverse direction, people who said they would retire because they could afford to had higher WRQL scores in every dimension by between 0.2 and 0.4 standard deviations. Social workers who said they would retire because they wanted to were similarly advantaged across the board, with Control at Work and Job and Career Satisfaction scores significantly higher after adjustment. Those who cited their occupational pension as a reason to retire also scored more highly on the Home–Work Interface. As these attitudinal indicators likely function as proxies for wealth, these findings conform with the extensive literature on socioeconomic gradients of health. It is nonetheless important to note that even within the social work profession with modest pay, these gradients are manifest and that having the leeway to retire at a chosen moment and to do so while mentally healthy is a further outworking of health inequalities within social work.

Social workers considering retirement to receive state benefits had a lower overall work-related quality of life, once other retirement perceptions are adjusted for. These individuals primarily felt lower control at work. WRQL was also lower among those who expected that their employer would not want them to work past a certain age, with this group registering significantly lower job and career satisfaction. Those who indicated they had other reasons to retire scored lower on all dimensions.

In terms of what employers could provide to incentivise extending work life, the majority of significant coefficients here are negative, suggesting that where respondents cited a provision they would favour, this is likely not currently on offer from their employer. An exception was that the social workers who said the option to work part-time would help them work longer had higher scores on all WRQL dimensions except for general wellbeing. This is robust to adjusting for other perceptions, including the ability to afford retirement, and suggests that people interested in continuing their work in a different pattern have a positive perception of their job. By contrast, those who express support for flexible work have lower scores across several dimensions, most pronounced with the dimension of Home–Work Interface, where scores are 0.5 SDs lower than others.

Those who favour retraining or updating skills as a means to prolong careers have lower scores, particularly on Job and Career Satisfaction. Retraining for a new role is associated with lower Control at Work and General Wellbeing scores, while wishing to update skills is associated with greater Stress at Work scores. Citing a less demanding role as a favoured provision is associated with greater Stress at Work and worse Working Conditions and General Wellbeing scores.

Overall it is clear that perceptions of career changes related to retirement are associated with levels of wellbeing and broader perceptions of working life, but that different retirement attitudes load onto distinct domains of Work-Related Quality of Life. 

### 3.2. Latent Class Results

We find evidence for an 8-class solution: AIC improves steadily from 4-class to 8-class versions and declines sharply from 9-class iterations onwards. A full profile of all 8 classes can be found in Appendix A. We acknowledge that while Hancock and colleagues draw special attention to AIC as the key innovation in LCA in recent decades [30], Bayesian information criterion and log likelihood scree plots are preferred by some analysts, and therefore, we have included these alongside the AIC plot in Appendix A. Bayesian Information Criterion (BIC) is similar for the 6- and 8-class solution, so we opted for the 8-class, which is the optimal point for AIC (our preferred index) and Log Likelihood. (A further LCA specification explicitly included the WRQL dimensions. However, class iterations did not show a smooth distribution of AIC, and the profiles suggested the classes were overdetermined in terms of wellbeing.)

There are three classes that correspond directly with responses to questions on retirement expectations, i.e., the age when and stage relative to the pension age when the person expects to retire. Class 5 identifies those who want to retire in their 50 s (necessarily before the anticipated pension age) —“Very Early Retirees”. Class 2 identifies others with plans to retire early, but not until their 60 s—“Early Retirees”. Class 7 segments those with low propensity to retire, citing either their 70 s as their planned retirement period, or not reporting any targeted retirement—“Late or Non-Retirees”.

All members of Classes 3 and 4 stated that they plan to rogetire at the age of pension entitlement in their 60 s. Responses that distinguish these two groups include support for named provisions to support extended working, reasons for retiring, and the likelihood of career change. Both had an above-average likelihood to cite state benefits and pension entitlements as reasons to retire, although this was further the case for Class 4, who were also more likely to ascribe retirement to ill-health and less likely to say they would retire because they could afford to. Class 4 had higher scores on the likelihood of mid-career job change and were much more likely to cite work stress or ill-health as reasons for a change. Both groups expressed support for some provisions to extend working life; however, they differ in support for retraining for a new role, updating skills, and taking on a less demanding role, with Class 4 much more likely to express interest in these options.

Overall, Class 3 and 4 have similar desired destinations, but Class 3 has a *straighter path* to retirement at pension age, whereas Class 4 alludes to more challenges in their journey—“Retire on Time: Straight Path” and “Retire on Time: Bumpy Road”.

Class 1 is heterogeneous in their retirement plans. The distinguishing feature of this group is that they support almost no provisions for extending working life, with less than 2% registering interest for any provision. They have an above-average likelihood of citing occupational or personal pensions as a reason to retire and a below-average likelihood of saying they would change career paths to vary their experience. Although the timing of their retirement is variable, they are “Retirement Inelastic”.

Class 6, the “Health-oriented” group, is distinct for being over 2.5 times more likely to cite ill-health as the reason for their retirement. They cite the highest likelihood of changing career paths and the vast majority of the group cite wellbeing reasons for considering such a change. Conversely, they are less likely than others to cite benefits or pensions as a reason for retirement. They give above-average support to every provision.

All those assigned to Class 8 stated that they would retire in their 60 s, although there was variability in whether this would be before, after, or at the point of pension entitlement. They have close to the average sample response on most items, although they are unlikely to cite stress or wellbeing as a reason for a change in career path or to cite benefits, affordability, or wanting to retire as reasons. Our interpretation is that this class model produces Class 8 as a median or unexceptional category who are “Retirement Agnostic”. There is no evidence that this group either prioritises or problematises the prospect of retirement, conforming to the mainstream conception of retirement as something which happens during one’s 60 s. It is distinct from the “Straight path” group in not having specificity around pensionable age or citing monetary benefits, so it has neither the advantages or priorities of this group nor the challenges of the “Health-oriented” group.

Although the Very Early, Early, and Late Retirees are mainly distinguished by the timing of when they plan to retire, they also share other observed characteristics used to create the classes. All were more likely to say they would consider a change of career path due to their job being stressful, both Early Retiree groups were more likely to cite effects on their wellbeing. Both Early Retiree groups were less likely to cite state benefits as a reason to retire. Occupational pensions were cited more often as a retirement reason by Early Retirees and less often by Late Retirees. Both Early Retiree groups were also more likely to cite affording to, wanting to, and retiring at the same time as their partner as reasons. Both were also less likely to opt for retraining or upskilling as a means of extending their working lives.

This class scheme is predictive of key variables not used in their creation. Work-Related Quality of Life scores are significantly lower for Health-oriented, Late/Non-Retirees, Very Early Retirees, and the Bumpy Road group members. For Late/Non-Retirees, this is driven primarily by differences in the Home–Work Interface dimensions. The other groups are closely bunched, although the Retirement Inelastic group has lower Stress at Work scores.

## 4. Discussion

The principal contribution of this paper is to underscore the complex interplay between perceptions of career and retirement, wellbeing, and quality of working life for health and social service workers. The analysis has pointed to the multitude of influence factors that affect individuals’ perceptions and plans regarding their late employment careers, in line with current career literature. Professional wellbeing and experiences of work are shown to differ between groups with differing attitudes towards why one would choose to retire from social work and what conditions would encourage extending one’s working life. Furthermore, different aspects of Work-Related Quality of Life measurements are associated with different attitudes: for example, Home-Work Interface scores are lower for people who favour flexible working arrangements as a means of extending working life, but higher for those who favour part-time work.

The theoretical prior we presented suggests the possibility of detecting seven latent classes through combining four simplified binary variables: retired/non-retired; retire fully or continue working; healthy/unhealthy; health status related to retirement status/unrelated. Although the 8-class result produced by the LCA procedure, summarized in Table 2, is similar in number to the prior, those classes do not straightforwardly map onto the theoretical prior.

The observed classes reveal diversity in orientation towards retirement and correlate with wellbeing. As in the prior, there are three classes with lower wellbeing, two of which more frequently cite their health as a reason they would retire. It is interesting to note the very high support among the group with the lowest wellbeing for all suggested workplace accommodations, including part-time and flexible work arrangements and retraining. This suggests that from the perspective of this group, a modified work arrangement could have health benefits, perhaps more so than immediate retirement, given the financial burden this would incur. People who continue in social work despite ill-health implicitly reveal a motivation to continue in work, be that vocational or financial. So this group better approximates the counterfactual where modified working conditions may make a difference to career sustainability, rather than one where ill-health is unaffected by retirement (a category we do not readily observe).

The Very Early Retiree category, who intends to retire in their 50 s, also reports lower wellbeing. They also support multiple employer provisions, but it is possible that some of this group will have improved outcomes when they retire, while others might benefit from an adapted work schedule.

The most valuable distinction produced by this LCA model is between two sets of social workers with similar plans to work until reaching pension entitlement, but differing levels of work quality of life, reasons for retiring, and interest in work accommodations. Those classified as on a “Bumpy Road” trajectory to retirement at the age of pension entitlement were more likely to be interested in changes of career path, had higher scores on the Intention to Leave scale, and more often cited stress and wellbeing as reasons to consider a change. Therefore, even though these individuals may not retire from work for some time, it is likely that they may leave the social work profession or the mode of service they have worked in to date. This may appear to HR managers and workforce planners as a potential in disruption to the workforce and loss of experience. However, open conversations with workers interested in career change could identify those factors that might make their current careers more sustainable and hence more desirable and/or could facilitate their movement into alternative areas of social work where their experience can be beneficial and not lost to the profession entirely.

There is tentative evidence that another underlying difference between these two groups and between those with firmer retirement plans and plans to retire in the medium term may map onto socioeconomic differences. Socioeconomic factors can influence both decisions around retirement planning [10,33] and wellbeing [34]. While our prior typology was deliberately simplified and naïve, it is clear that the omission of socioeconomic circumstances leaves an incomplete picture. A weakness of the SWAS dataset is that we did not collect a good proxy indicator for socioeconomic status outside of attitudes to retirement, such as being able to afford to retire. Future research could do more to differentiate between people whose orientation towards retirement derives primarily from financial security versus those enticed by other benefits.

It is noteworthy that these data point to a substantial quotient of the social worker population who are seemingly not engaged with the subject or prospect of retirement. This points to a potential dividend from increasing the level of education and information offered to social workers on choices facing them towards the end of their careers. Furthermore, social workers demonstrate variable levels of enthusiasm for the options of training, reskilling and role transition, which are seen differently by different groups. It may be that there are variable impressions of what new training would entail and for what one would be retraining. Underlying this may be a spectrum of opinion on the concept of “age-appropriate roles”, such as mentoring, clinical supervision and governance. There are risks to exposing undertrained staff to unfamiliar roles, but there may also be untapped potential among social workers who have been given insufficient clarity on the training and transitions available to them.

## 5. Further Limitations

This study puts a microscope on conceptions of retirement and their relationship with health. Health and wellbeing exist within a very wide array of factors, and further work is needed to contextualise the importance of career and retirement prospects relative to these other factors. For example, poor working conditions can contribute both to wellbeing and to a desire to retire or change jobs. Furthermore, this study does not attempt to disentangle the two-way relationship between one’s health and how one sees their work, career, and the prospect of retirement. Identifying the precise causal impact of changing a person’s approach to retirement would require a more involved strategy, which may be required as this field of study progresses. However, demonstrating the complex associations between wellbeing and retirement attitudes is of value in the interim.

The survey data on hand had much less coverage of retired people. By nature, they are a smaller population and also harder to access through professional publications. Further research is needed to understand retirees’ wellbeing and the extent to which they regret leaving the social work profession. In addition to socioeconomic information, there are other important factors that could further contextualise attitudes to retirement, including grandparenting responsibilities.

This study represents a first exploratory attempt to synthesise a range of perceptions of retirement into meaningful groupings. Three of the eight classes were uniform in their responses to questions on when they anticipated retiring. The fact that these items were discrete choice and forced respondents to choose between options, as opposed to the “tick all that apply” items on retirement reasons and provisions, may have resulted in these items having an outsized effect on the constitution of the classes. Nonetheless, the inclusion of these other items has clearly added to the discriminant power of the model and has helped to distinguish individuals with similar retirement plans but distinct perceptions of work and retirement more broadly. It is unclear whether these classes reflect specific challenges to social workers and whether other categories of HSS workers or other occupations would divide into similar groupings. It may also be possible that because the underlying questions are quite general, the resulting classes may lack job-oriented specificity or nuance, which would be invaluable for planners. The challenge is to find and accrue similarly amenable data for other groups of workers so as to test the generalisability of the scheme we propose.

## 6. Conclusions

In the period in which this research took place, the Northern Ireland government published the *Power to People* report [2], setting out the policy direction and challenges for the provision of social care. Maintaining a sufficient workforce is central to that strategy, but predicting the size of the workforce available in a given year or day is highly complex. It is clear that inaction on the related issues of flexible provisions for a late career stage and of dissatisfaction with work comes at a cost. From a planning perspective, each represents a route through which people can leave the workforce and a source of uncertainty.

The direction of travel in social policy has been towards what Macnichol refers to as “neoliberalising old age” [4], i.e., viewing older people as an underused labour resource. Taken as the whole, this study underscores that the unintended consequences of too broad of a move to keep social workers employed will include disruption to people’s life plans and likely entail costs to individual wellbeing. This could, in turn, incur a hidden cost on service capacity and increase the long-run demand for the same health and social services, whose delivery is at issue.

Furthermore, we demonstrate that it is difficult to straightforwardly identify groups who are both amenable to changing their retirement plans and who would benefit from any policy innovation. The identification of a “Retirement Inelastic” group, whose retirement plans are not open to revision through any of the suggested employer accommodations, is a reminder that any innovation is unlikely to achieve universal uptake.

Finally, the current COVID-19 pandemic has brought the issue of retirement from health and social services to greater prominence. Many frontline workers have returned from retirement to contribute to frontline responses to the pandemic, either through formal recruitment campaigns or voluntary schemes [34,35]. This underlines the skill and expertise that people are capable of bringing to services in the years beyond which they would have anticipated working. This may prompt individuals to reassess whether they are likely to be valued by the system in which they are employed beyond the traditional normative retirement age. It may lead to longer-term reassessment from a governmental perspective of how many good people are currently “lost” to service provisions either through a preference for retirement or insufficient knowledge of how they can continue to contribute to the workforce beyond pension entitlement. Given what this paper has shown about the attachment some people demonstrate to their long-term retirement plans, we would urge caution and sensitivity to individual circumstances when considering moves to alter workers’ incentives and entitlements to retire.

## Figures and Tables

**Table 1 ijerph-17-04722-t001:** Descriptive statistics.

	Frequency/Mean	% / Standard Deviation
*Reasons for Retirement*		
To receive state benefits	221	18.23%
To receive occupational or personal pension	328	27.06%
To retire same time as partner	72	5.94%
Can afford to	227	18.73%
Want to	478	39.44%
Ill-health	103	8.50%
Don’t expect employer will want me to work on	100	8.25%
Other reason	219	18.05%
Number of Retirement Reasons	1.44	1.24
*Provisions*		
Retrain for new role	325	26.79%
Course to update skills	273	22.52%
Flexible hours option	667	54.99%
Part-time	657	54.21%
Less demanding role	510	42.08%
Longer holidays	531	43.78%
Long break option	587	48.39%
Number of Favoured Provisions	2.93	1.98
None of these	133	10.97%
Career Move Likelihood (Likert, 1–5)	2.71	1.40
*Reasons for Career Path Change*		
Variety of practice experience	139	11.43%
Job stressful	292	23.74%
Job impacted health and wellbeing	399	32.49%
Just wanted a change	80	6.57%
Other reason	221	18.11%
*When Planning to Retire*		
Before 60	152	12.53%
60 s	820	67.6%
After 70 s	88	7.25%
Not Applicable	153	12.61%
Before anticipated pension age	453	37.47%
At anticipated pension age	448	37.06%
After my anticipated pension age	128	10.59%
Not planning to retire	33	2.73%
Retired already	70	5.81%
Don’t know	142	11.75%
How likely is it that your organisation would consider a flexible working request? (Likert, 1–5)	3.46	1.42
Intention to Leave Scale	42.09	8.70
*Work-Related Quality of Life*		
Job and Career Satisfaction	20.12	4.57
Stress at Work	7.59	1.88
Work Conditions	9.35	2.43
Control at Work	9.47	2.75
General Wellbeing	19.23	4.67
Home-Work Interface	9.58	2.91
Total WRQoL	72.17	15.17

**Table 2 ijerph-17-04722-t002:** Latent Class Descriptions and Frequencies.

Class/Frequency	Description
Late Retirees (7; *n* = 167) Retirement Agnostic (8; *n* = 179) Pension Age Retirement—Smooth Path (3; *n* = 190) Early Retirees (2; *n* = 302) Very Early Retirees (5; *n* = 106) Pension Age Retirement—Bumpy Road (4; *n* = 190) Health-oriented (6; *n* = 189) Retirement Inelastic (1; *n* = 161)	Healthy, in Work—Would be Unhealthy if Retired Healthy, in Work—Would be the Same if Retired Completely Healthy, in Work—Would be the Same if Career Concluded, but Work Continued Unhealthy, in Work—Would be Healthy if Retired Completely Unhealthy, in Work—Would be Healthy if Career Concluded, but Work Continued Unhealthy, in Work—Would be the Same/Unhealthy if Retired Retired

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
