# Peer review of "Seeing the Finish Line? Retirement Perceptions and Wellbeing among Social Workers"

_ijerph, 2020, doi:10.3390/ijerph17134722_

Round 1
Reviewer 1 Report
Population aging has been poised as one of the most significant social transformations, including labor and financial markets, and the demands for goods and services. Many countries are likely to experience fiscal and political pressures in relation to public systems of health care, pensions, and social protections for a growing older population. Employment and health are inextricably linked. Understanding the retirement perceptions may inform the policies making to extend working life and enhance the work-related quality of life. This manuscript explored the retirement perceptions and wellbeing among social works that can be of interest to the International Journal of Environmental Research and Public Health readers. I would like to humbly suggest some revisions before publishing this manuscript.
The introduction delineated the complexity of retirement perceptions and the experiences among the aging population in the workplace. Since this manuscript has an emphasis on the social worker profession, having an overall description of the social workers in the UK can narrow the focus for readers internationally. In addition, a brief introduction of current policies in the UK can inform explicit directions and discussion points.
Several major issues in the methods and results section are in need of revision. It appears missing information for the latent class analysis description (p.3, line 129). Authors alleged seven subscales for Work-Related Quality of Life (WRQL), but presented 6 subscales in the text and the supplementary File A. In addition, I encourage authors to reorganize the methods section to eliminate confusion for readers. Having a section to illustrate measures and another section for data analysis can be one approach. Specifically, I would suggest merging the social attitude survey from (p. 3, line 108-120 & p. 4 line 160-170), as both appeared to be the same survey and adapted from the British Attitudes Survey. Please be clear and report if those are two different social attitude surveys, as one specific indicated 2015, the other does not.
The authors explained the prior reasoning and broke the current workforces down to six categories. I wonder if there is any existing theoretical model that can support the authors’ rationale. It appeared to me authors assumed the causality of health and work-related satisfaction. Another aim in this manuscript is the transition, maybe authors can consider Super career development model as a way to conceptualize the life span transition.
In addition, it is unclear how authors came to table 1 and the 8 class, there are Information Criterion (ICs) used to determine the number classes, Bayesian Information Criterion (BIC) performed the best of the ICs and the bootstrap likelihood ratio test have proved to be a very consistent indicator of classes. Suggesting in need of including those ICs and how to interpret the ICs. With those ICs the results can be interpreted in a more meaningful way and to inform the discussion and implication.
Lastly, the overall manuscript should be more consistent and make supplement files reader-friendly (For example, simply provide column and row name to the reader, q21_w~50 is hard for the reader to understand).
Author Response
Thank you for your thorough and constructive comments and for encouraging us to improve our manuscript.
- Supplementary file/presentation of measures: The variables you highlighted are now clearly labeled and readers should be able to make a clear connection to the text descriptions. The methods section is divided along the lines that you suggest with measures consolidated into a single section and referred to again briefly when the analysis is outlined.
- Latent class: AICs were adverted to in the original version and these were visualised in Supplementary file B. Though this is our preferred indicator, we acknowledge that BIC and LL are commonly used measures preferred by other authors so have included these scree plots in the supplementary file and referred to this in a footnote. These do not challenge our selection of the 8-class solution.
- Theoretical justification: We thank you for highlighting this as an area for development, and another reviewer was in agreement. While we try to avoid a ‘age determines stage’ approach, we acknowledge that Super's career development model is an important contribution to an ever-evolving literature. You will find our introduction anchors the paper in a framework of career sustainability and self-concept, which we refer back to directly when introducing the theorised priors in the methodology and again in the discussion. The paper is greatly strengthened for having this theoretical emphasis.
- You are correct that there are six subscales to Work Quality of Life: the total makes a seventh relevant measure, but this was previously worded incorrectly and is now amended, as is the incomplete sentence you identified in the methodology section.
Reviewer 2 Report
Dear Authors,
Thank you for the opportunity of reviewing this paper, concerning the “Retirement Perceptions and Wellbeing among Social Workers”. This is a paper, which captures the attention of the reader.
Given the sociodemographic changes and challenges for ageing in Western populations, the theme is extremely relevant for HSS professionals (and health professionals in general), and constitutes a substantive contribution for occupational health investigation and ageing, showing that the process of human adaptation to stress and change is interactive and continuous, affecting the person’s wellbeing and work-related quality of Life.
The manuscript is theoretically logical, parsimonious, useful and well documented. The methodological processes are appropriate, clear and well grounded. The results and the conclusion sections, reveal a comprehensive style, linking the purpose of the study with theory and the main implications for practice. The limitations of the study are well stablished. Even though, some minor aspects should be addressed:
- In the abstract, lines 21 to 27, the authors summarize the two main steps of the research design, although that this section should clearly highlight the novelty and the main results of the study;
- In methods section, line 84, authors sate that “The survey was hosted by Ulster University and promoted by Community Care (www.communitycare.co.uk)...”; but it is not clear if the research addressed the necessary ethical standards;
- In order to increase the validity of the results, in the methods section, authors should consider to present the psychometric characteristics for each instrument used;
- The discussion section could be improved by linking the purpose of the study with theory.
Best regards
Author Response
Thank you very much for your encouragement and for your very helpful comments:
- Abstract: Having re-read this section of the abstract, I agree that the original wording was insufficiently direct and have sought to emphasise the findings more clearly in the revised manuscript.
- Ethics:Thank you for pointing out this oversight. A paragraph outlining the ethical stipulations and approval for this study has been included.
- Published Cronbach's Alpha scores for WRQL and ITL scales are now included
- Thank you for highlighting the need for greater theoretical linkage, a point on which another reviewer agreed. In the introduction and methods and discussion, we have appealed to contemporary theories of career sustainability and self-concept.
Reviewer 3 Report
This is a very timely and quite important and novel research on retirement. Well done. The questions proposed are quite appropriate and important. The methodology is complex but so is the subject of inquiry. Policy makers beware: "Given what this paper has shown about the attachment some people demonstrate to their long-term retirement plans, we urge caution and sensitivity to individual circumstances when considering moves to alter worker's incentives and entitlements to retire." Now we have so valid support to make that statement. Thank you!
Author Response
Thank you for reading our article and for your encouragement.